# Increasing Electrical Resistivity of P-Type BiFeO_3_ Ceramics by Hydrogen Peroxide-Assisted Hydrothermal Synthesis

**DOI:** 10.3390/ma16083130

**Published:** 2023-04-16

**Authors:** Cristian Casut, Raul Bucur, Daniel Ursu, Iosif Malaescu, Marinela Miclau

**Affiliations:** 1National Institute for Research and Development in Electrochemistry and Condensed Matter, Plautius Andronescu Str., No. 1, 300224 Timisoara, Romania; 2Physics Faculty, West University of Timisoara, V. Pârvan Ave., No. 4, 300223 Timisoara, Romania; 3Institute for Advanced Environmental Research, West University of Timisoara (ICAM-WUT), Oituz Str., No. 4, 300086 Timisoara, Romania

**Keywords:** bismuth ferrite, hydrothermal synthesis, vacancy, hydrogen peroxide, complex dielectric permittivity

## Abstract

Bismuth ferrite (BiFeO_3_, BFO) is still widely investigated both because of the great diversity of its possible applications and from the perspective of intrinsic defect engineering in the perovskite structure. Defect control in BiFeO_3_ semiconductors could provide a key technology for overcoming undesirable limitations, namely, a strong leakage current, which is attributed to the presence of oxygen vacancies (*V_O_*) and Bi vacancies (*V_Bi_*). Our study proposes a hydrothermal method for the reduction of the concentration of *V_Bi_* during the ceramic synthesis of BiFeO_3_.Using hydrogen peroxide (H_2_O_2_) as part of the medium, *p-type* BiFeO_3_ ceramics characterized by their low conductivity were obtained. Hydrogen peroxide acted as the electron donor in the perovskite structure, controlling *V_Bi_* in the BiFeO_3_ semiconductor, which caused the dielectric constant and loss to decrease along with the electrical resistivity. The reduction of Bi vacancies highlighted by a FT-IR and Mott—Schottky analysis has an expected contribution to the dielectric characteristic. A decrease in the dielectric constant (with approximately 40%) and loss (3 times) and an increase of the electrical resistivity (by 3 times) was achieved by the hydrogen peroxide-assisted hydrothermal synthesized BFO ceramics, as compared with the hydrothermal synthesized BFOs.

## 1. Introduction

Bismuth ferrite (BiFeO_3_, BFO) is being widely investigated because of the great diversity of its possible applications that offer a wide range of potentially new applications including spintronics [1], new data storage media [2], multiple-state memories [3,4], ferroelectric diode devices [5], ferroelectric photovoltaics [6,7] or as a photoelectrode in a solar water-splitting cell [8].

However, its commercialization is conditioned by the tendency to exhibit strong leakage currents, which have often been attributed to the presence of oxygen vacancies (*V_O_*) [9,10] and Bi vacancies (*V_Bi_*) [11]. If oxygen vacancies dominate the conductivity of BFO under oxygen-poor conditions, determining *n-type* behavior [12,13,14,15], Bi vacancies play a dominant role under oxygen-rich conditions, causing *p-type* conductivity of BFO [16]. 

Besides its potential applications, the defect control in BiFeO_3_ semiconductors is also interesting and fascinating and could provide a key technology for overcoming undesirable limitations. So far, many papers have mainly focused on studying the oxygen vacancies and their reduction in correlation with the properties as the main defects in BFO thin films and ceramics [17,18]. It has been reported that oxygen vacancies can be reduced through doping or treatment in an oxygen atmosphere; thus, an increase in the effective resistance has been highlighted [19,20]. From this perspective, using hydrogen peroxide as oxygen sources in the solutions was explored for the reduction of the concentration of oxygen vacancies presented in *n-type* BFO [21].

However, few studies on *V_Bi_* in BFO thin films and ceramics have been published, especially from the perspective of first-principles density functional calculations [22,23]. In accordance with theoretical calculations, the most stable state for V_Bi_ is the fully ionized one, V_Bi_^3−^, obtained by release of their holes, which determined the *p-type* conduction. Thus, BiFeO_3_ acts as a *p-type* semiconductor with a high concentration of h^+^. As a result, the conductivity of BFO can be reduced by decreasing the concentration of V_Bi_ through two mechanisms, namely, increasing the donor concentration or decreasing the acceptor concentration. To our knowledge, one study reported that the conductivity of *p-type* BFO decreased through the reduction of the concentration of *V_B_*_i_ due to the doping of BFO nanofibers with Sn by the sol–gel electrospinning technique [16]. 

In this paper, we propose a reduction of the concentration of *V_B_*_i_ during the synthesis of BFO ceramics. Thus, using hydrogen peroxide (H_2_O_2_) as part of a hydrothermal medium, *p-type* BiFeO_3_ ceramics characterized by low conductivity have been obtained. In accordance with our study, hydrogen peroxide acted as an electron donor in the perovskite structure, controlling the defects in the BiFeO_3_ semiconductor.

## 2. Materials and Methods

In a typical hydrothermal synthesis described in our earlier work [24], we were able to produce BiFeO_3_ (BFO). Sample 1 (S1), which will be used as a reference, was obtained by separately mixing, 1 mmol (0.5 g) of bismuth nitrate (Bi(NO_3_)_3_ × 5H_2_O ≥ 98%) and 1 mmol (0.4 g) of ferric nitrate (Fe(NO_3_)_3_ × 9H_2_O ≥ 98%) in 5 mL distilled water. Both solutions were then homogenized for 15 min using magnetic stirring, generating a brownish-yellow solution. The sample mixture was combined with a 10 mL of 1 M sodium hydroxide (Na (OH) ≥ 99%) solution before being transferred to a 60 mL Teflon line autoclave and then heated at 200 °C for 12 h.

By substituting 5 mL of water from the bespoke approach with H_2_O_2_, we propose a novel one-step hydrothermal method for controlling the vacancies. The as-synthesized BFO will be further noted with S3.

Another sample (labeled S2) was prepared by adding a very small amount of bismuth nitrate, 0.1 mmol (0.05 g) more into an identical synthesis procedure as S1. 

An XRD PANalytical X’Pert PROMPD Diffractometer (Almelo, The Netherlands) was used to analyze and identify the samples at room temperature. Using Cu Ka radiation at 40 kV and 30 mA, the XRD characterizations were conducted across the scanning range of 10° to 80°. 

Scanning Electron Microscopy (SEM/EDX, Inspect S model, Eindhoven, The Netherlands) was used to examine the morphology and microstructure of the samples.

Fourier transform infrared (FT-IR) spectra were acquired on a JASCO-430 Fourier transform spectrometer (Jasco Inc., Tokyo, Japan) using the KBr pellet technique with resolutions ranging from 2000 to 400 cm^−1^. 

Using a three-electrode cell composed of a 0.28 cm^2^ BFO film as the working electrode and a 0.28 cm^2^ Ag/AgCl/KCl satellite electrode linked to a Luggin capillary as the reference electrode, electrochemical investigations were carried out using a potentiostat model PGZ 402 (Voltalab, France).

After being combined with a binder solution of polyvinyl alcohol (5% PVA), all three samples were pressed into disks of the same size (6 mm in diameter and approximately 1 mm thick) for the dielectric measurements. These disks were weighted and measured, and densities were determined using Archimedes’ technique, after sintering. The ceramics’ relative densities were calculated as a percentage of the theoretical density [25].

A coating of Ag was placed on both polished surfaces of the disks, representing the two electrodes that were connected to an LCR meter. Complex impedance measurements were performed at various frequencies between 100 Hz and 2 kHz to establish the frequency dependence of the dielectric constant, dielectric loss, and electrical resistance.

## 3. Results and Discussion

Figure 1 shows the X-ray diffraction pattern of BFO powders prepared using the hydrothermal technique. BiFeO_3_ (JCPDS no. 01-072-2321) with a rhombohedral structure is indexed for all diffraction peaks of S1 not highlighting the development of the impurity phases during the synthesis procedure. Both samples S2 and S3 exhibited the development of Bi_25_FeO_40_ impurity phases. The excess oxygen provided by hydrogen peroxide stabilized the Bi_25_FeO_40_ phase in S3. Using the XRD data, the levels of the impurity phases were roughly quantified, with an impurity content of 13% for both. S2, characterized by the same concentration of the impurity phase of Bi_25_FeO_40_ as S3, was used as a standard to exclude the effect of this phase on the structural or electrical characteristics of BFO, highlighting only the effect of hydrogen peroxide on the concentration of vacancies in the BFO structure. According to the first-principles density functional theory calculations [11], the presence of Bi vacancies will change the structural parameters and will be responsible for crystal volume reduction. The quantitative XRD data analysis results using Rietveld refinement (Figure 2) with X’Pert HighScore Plus revealed that the crystal volume of S3 (373.75 Å) is higher than S1 (373.39 Å) and can be correlated with the reduction of Bi vacancies by using hydrogen peroxide in the hydrothermal synthesis.

SEM scans (Figure 3a–c) reveal that the resultant BFOs are large-scale aggregations of truncated and highly deformed polyhedra with an average edge length of roughly 10 µm. Moreover, no difference in the size or morphology of the produced BFO ceramics by a hydrothermal route with and without hydrogen peroxide was observed.

Table 1 shows the crystalline phases of the final products, as well as the key synthesis parameters that were modified from the standard technique.

In contrast to the XRD analysis, a FTIR analysis is much more sensitive to the presence of defects, providing direct information about the modification of the structure of BFO ceramics due to vacancies. The FT-IR spectra of the samples is presented in Figure 4. For all samples, the bands centered at 1640 cm^−1^ correspond to the O–H stretching modes of interlayer water molecules and the bending mode of water molecules δ(H_2_O), respectively [26,27]. 

From the viewpoint of *V_Bi_,* the region of 650–400 cm^−1^, corresponding to the vibration modes of the FeO_6_ octahedron, is interesting for analysis. Each sample highlights the two modes of BFO caused by the O-Fe-O bending vibration, *E(TO8)* and Fe−O stretching vibration, *E(TO9)*. In addition, the absorption peak of S2 and S3 detected at 572 cm^−1^ is the vibrational “fingerprint” of Bi_25_FeO_40_ and corresponds to the stretching vibration of the Fe−O bond [28]. It can be seen that the O-Fe-O bending vibration of BFO is not affected by hydrogen peroxide or the Bi_25_FeO_40_ phase, being identifiable at approximately 452 cm^−1^ for all three BFOs (Figure 5a). 

On the contrary, the stretching vibration of the Fe–O bond of BFO changed from 555 cm^−1^ for S1 and S2, respectively, to 538 cm^−1^ for S3 (Figure 5b). From the first-principles calculations, compared to perfect structures, the presence of a Bi vacancy reduces the Fe-O bond [29].

The bond lengths (r) for all the samples were calculated from the values of the force constant (*k*), using Equations (1) and (2) [28].
(1)$$f=\frac{1}{2\pi }\sqrt{\frac{k}{µ}}$$
(2)$$k=\frac{17}{r^{3}}$$
where *f* is the vibration frequency and µ is the effective mass (2.0644 × 10^−26^ kg).

The interatomic bond length, Fe-O, is much longer in S3 (2 Å) than in S1 or S2 (≈1.96 Å) (Table 2), confirming the reduction of *V_Bi_* in the sample synthesized by the hydrogen peroxide-assisted hydrothermal method.

Moreover, for all three samples, the Mott–Schottky analysis of the BiFeO_3_ phase showed a negative slope, indicating *p-type* conductivity. S1, synthesized without H_2_O_2_, was characterized in our previous work [24]. The *p-type* semiconductor behavior is a characteristic of BFO synthesized by the hydrothermal method in an oxidizing medium, generating Bi vacancies, which, by ionization, release their holes [16]. In addition, the *n-type* conductivity specific to Bi_25_FeO_40_ is highlighted by the Mott–Schottky analysis of S3 (Figure 6) [30].

For highlighting the number of *h^+^* responsible by the *p-type* conduction of the BFO phase, the Mott–Schottky equation [31] has been used to calculate the acceptor density (N_A_) from the slope of the linear region, considering that the dielectric constant of BFO material is 120 in the following equation [32]:$$N_{A}=-\frac{2}{{e\varepsilon \varepsilon }_{0}}{\left[\frac{d\left(C^{-2}\right)}{\mathrm{dV}}\right]}^{-1}$$
where C is defined as the capacitance of the space charge region, ε_0_ and ε are the vacuum permittivity and dielectric constant of BFO, respectively, e is the electron charge and V is the electrode applied potential. The hole density value of S1 is 7.57 × 10^17^ cm^−3^ and for S3 this is reduced to 3.49 × 10^17^ cm^−3^, confirming the beneficial effect of hydrogen peroxide on defect control in BiFeO_3_ semiconductors, even in the case of Bi vacancies.

The characteristics of BFO ceramics are heavily influenced by factors such as grain size and density. The observed results suggest that the experimental conditions used in the synthesis are suitable for the production of highly dense ceramics. Sample 2 has the highest degree of densification (relative density of 93%), while samples 1 and 3 have comparable values (≈92%). The high relative densities of all samples rule out any effect of disk preparation on the electrical studies.

Figure 7 depicts the frequency dependence of the dielectric constant (ε) and loss (tan δ) of the samples. It can be seen that both the dielectric constant (ε) and loss (tanδ) of S3 are lower than that of the samples prepared without hydrogen peroxide.

The values of the dielectric constant of S3 are smaller with approximately 40%, while the dielectric loss has an even more substantial decrement being more than 3 times smaller than those of the other two samples. S1 and S2 also have a much stronger dependence on frequency for the dielectric loss than S3, indicating that the leakage current had a significant influence on the dielectric properties of the first two [33]. The very low dielectric loss of S3 is in concordance with the evolution of electrical resistivity, which for S3 is more than three times larger than the reference values of S1 and S2 at 100 Hz.

These very low values of dielectric loss are in agreement with the values presented in Figure 8, where it can be seen that the electrical resistance of Sample 3 is more than three times larger at 100 Hz than the reference values of Samples 1 and 2.

The reduction of Bi vacancies highlighted by the FT-IR and Mott–Schottky analysis has an expected contribution to the dielectric characteristic, decreasing the dielectric constant and loss together with an increase of the electrical resistivity achieved by the H_2_O_2_-synthesised BFO ceramic. Furthermore, the similar dielectric behaviors of S1 and S2 reveal that the presence of the Bi_25_FeO_40_ parasitic phase does not significantly affect dielectric properties.

Taking into consideration the above results, a possible explanation is that hydrogen peroxide (H_2_O_2_) acted as the electron donor as in the following:H_2_O_2_(aq) → O_2_(g) + 2H^+^(aq) + 2e^−^


The electrons provided by the donor H_2_O_2_ compensate the h^+^ generated by the unoccupied acceptor *V_Bi_*, effectively diminishing the concentration of h^+^ and, therefore, leading to an increase in electrical resistivity. In addition, excess oxygen provided by hydrogen peroxide stabilized the Bi_25_FeO_40_ phase in S3.

## 4. Conclusions

Using hydrogen peroxide (H_2_O_2_) as part of a hydrothermal medium, *p-type* BiFeO_3_ ceramics characterized by high electrical resistivity have been obtained. FT-IR and Mott–Schottky analyses highlighted the reduction of *V_Bi_* in the sample synthesized by the hydrogen peroxide-assisted hydrothermal method. Thus, the interatomic bond length, Fe-O, is much longer in S3 (2 Å) than in S1 or S2 (1.96 Å). This, together with the reduction of the hole density value to 7.57 × 10^17^ cm^−3^ in the case of S1 and to 3.49 × 10^17^ cm^−3^ in the case of S3, confirms the beneficial effect of hydrogen peroxide on the defect control in BiFeO_3_ semiconductors, even in the case of Bi vacancies. The reduction of Bi vacancies has an expected contribution to the dielectric characteristic, namely decreasing the dielectric constant and loss, together with an increase of electrical resistivity. Furthermore, the similar dielectric behaviors of S1 and S2 reveal that the presence of the Bi_25_FeO_40_ parasitic phase does not significantly affect the dielectric properties. In conclusion, hydrogen peroxide (H_2_O_2_) acted as the electron donor. The electrons provided by the donor H_2_O_2_ compensated the h^+^ generated by the unoccupied acceptor *V_Bi_*, leading to a decrease in the concentration of h^+^. Enhancing the dielectric and electrical properties achieved by the H_2_O_2_-synthesised BFO ceramic confirms that bismuth vacancies can be reduced by this method. In our future work, we will study the effect of the amount of hydrogen peroxide on Bi vacancies, reflecting on the dielectric and electrical properties. In addition, the hydrogen peroxide-assisted hydrothermal synthesis might be extended to the synthesis of other perovskite materials.

## Figures and Tables

**Figure 1 materials-16-03130-f001:**
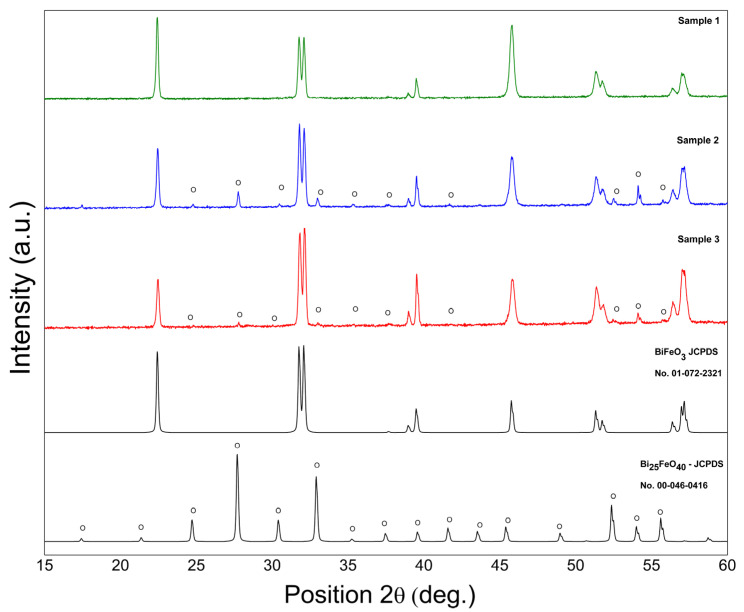
Room temperature X-ray diffraction pattern of BFO powders.

**Figure 2 materials-16-03130-f002:**
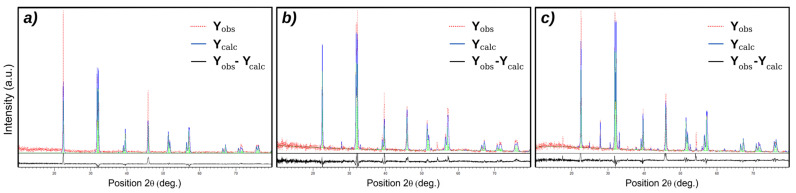
Rietveld refinement of the (**a**) Sample 1, (**b**) Sample 2 and (**c**) Sample 3.

**Figure 3 materials-16-03130-f003:**
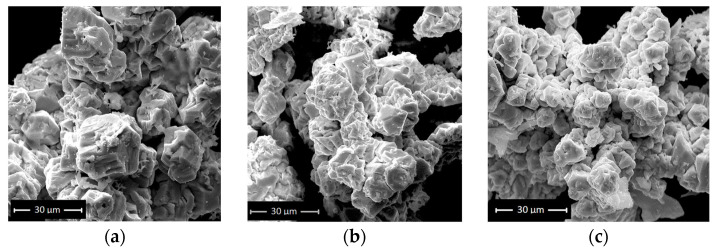
SEM image of (**a**) sample 1, (**b**) sample 2 and (**c**) sample 3.

**Figure 4 materials-16-03130-f004:**
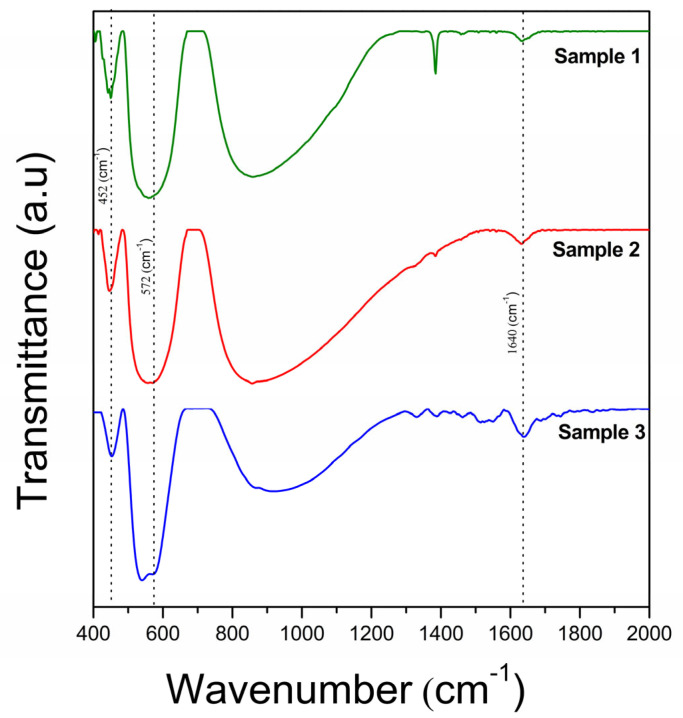
FT−IR spectra of the samples in the range of 400−2000 cm^−1^.

**Figure 5 materials-16-03130-f005:**
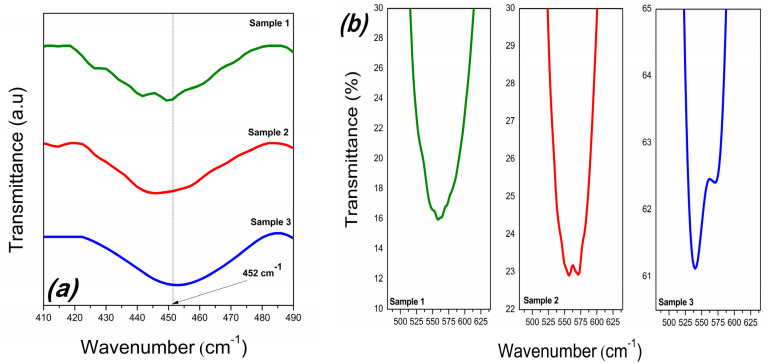
FT−IR spectra of the samples in the range of 400−490 cm^−1^ (**a**) and in the range of 480−640 cm^−1^ (**b**).

**Figure 6 materials-16-03130-f006:**
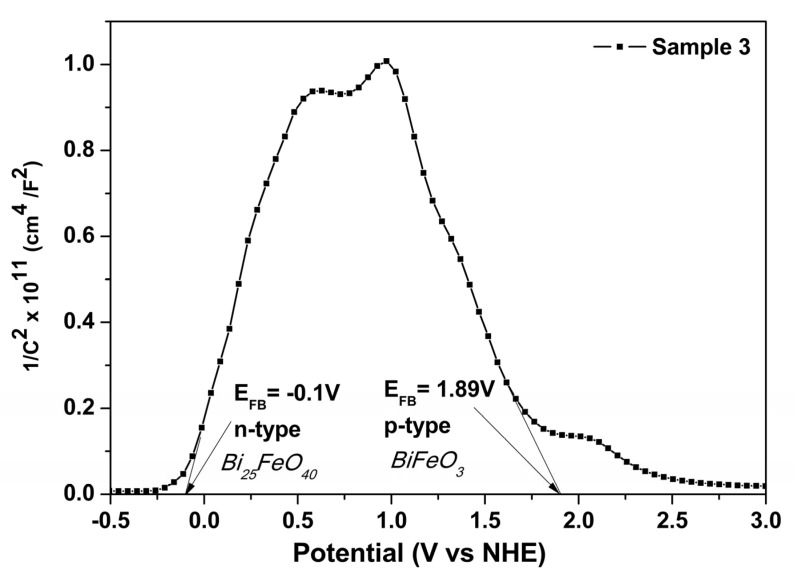
Mott−Schottky plot of Sample 3.

**Figure 7 materials-16-03130-f007:**
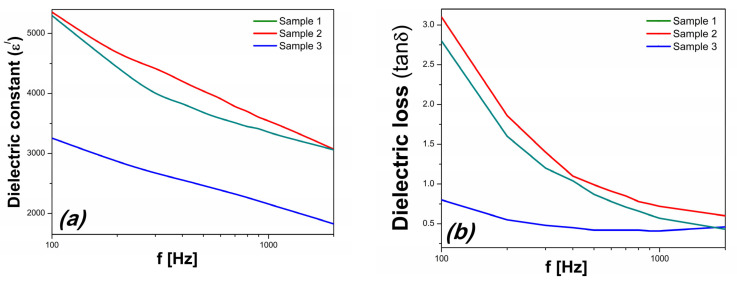
The frequency dependence of (**a**) the dielectric constant and (**b**) dielectric loss.

**Figure 8 materials-16-03130-f008:**
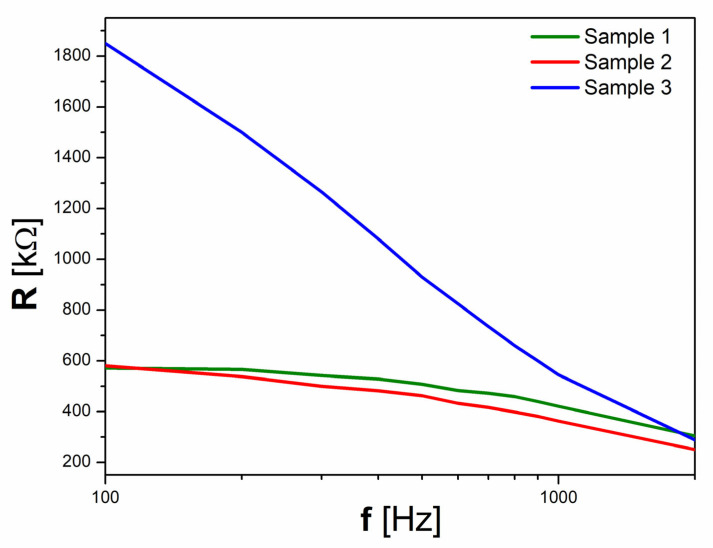
The frequency dependence of electrical resistance.

**Table 1 materials-16-03130-t001:** Crystalline phases of the final products and some synthesis parameters.

Product	Procentage of BiFeO_3_/Bi_25_FeO_40_	Quantity of H_2_O_2_/H_2_O (mL)	Steering in Closed Conditions	BiFeO_3_ Unit Cell Parameters (Å)
S1	100 (%)/0 (%)	0/15	No	5.576 (7) 5.576 (7) 13.864 (2)
S2	87 (%)/13 (%)	0/15	No	5.578 (3) 5.578 (3) 13.865 (7)
S3	87 (%)/13 (%)	5/10	Yes	5.579 (2) 5.579 (2) 13.867 (6)

**Table 2 materials-16-03130-t002:** Fe-O bond length for all samples.

	Wave Number ν (cm^−1^)	Force Constant k (N/cm)	Bond Length R (Å)
Sample 1	556	2.243	1.957
Sample 2	555	2.259	1.959
Sample 3	538	2.123	2

## Data Availability

The data presented in this study are available upon request from the corresponding author.

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
