# Peer review of "Increasing Electrical Resistivity of P-Type BiFeO_3_ Ceramics by Hydrogen Peroxide-Assisted Hydrothermal Synthesis"

_materials, 2023, doi:10.3390/ma16083130_

Round 1
Reviewer 1 Report
In this paper, the hydrothermal method of hydrogen peroxide to improve the resistivity of BiFeO3 ceramics was studied in detail. The topic is very interesting. The overall quality meets the journal standards, but it still needs to be partially revised. Therefore, the author suggests a slight modification.
In order to improve the clarity and quality of the manuscript, the following necessary modifications are recommended:
(1) In the introduction part, please introduce the mechanism of the effect of VBi on the performance of BiFeO3, why the change of VBi value will affect the performance of BiFeO3.
(2) The value of transmittance in Fig.3 ( b ) is not reflected in the figure, so it is recommended to supply.
(3) The number of test groups is not enough, and three groups are recommended to reduce the error of the test process under the same conditions.
(4) Figure 5 and Figure 6 are not clear enough so it is recommended to optimize.
Author Response
Thank you very much for the comments. We appreciate the referee’s time and effort to improve the manuscript. We revised the manuscript according to the referee’s comments. We hope that our responses to the comments and the changes in the text will be accepted by you.
Reviewer 1. In this paper, the hydrothermal method of hydrogen peroxide to improve the resistivity of BiFeO3 ceramics was studied in detail. The topic is very interesting. The overall quality meets the journal standards, but it still needs to be partially revised. Therefore, the author suggests a slight modification.
In order to improve the clarity and quality of the manuscript, the following necessary modifications are recommended:
- In the introduction part, please introduce the mechanism of the effect of VBi on the performance of BiFeO3, why the change of VBivalue will affect the performance of BiFeO3.
R1. However, few studies on VBi in BFO thin films and ceramics have been published, especially from the perspective of first-principles density functional calculations [19,20]. In accordance with the theoretical calculations, the most stable state for VBi is the fully ionized one, VBi3−, obtained by release of their holes, which determined the p-type conduction. Thus, BiFeO3 acts as a p-type semiconductor with a high concentration of h+. As a result, the conductivity of BFO can be reduced by decreasing the concentration of VBi through two mechanisms, namely increasing the donor concentration or decreasing the acceptor concentration. To our knowledge, one study reported that the conductivity of p-type BFO decreased through the reduction of the concentration of VBi due to the doping of BFO nanofibers with Sn by the sol–gel electrospinning technique [13].
- The value of transmittance in Fig.3 ( b ) is not reflected in the figure, so it is recommended to supply.
R2. We changed Fig. 3b to correspond to the reviewer’s recommendation.
- The number of test groups is not enough, and three groups are recommended to reduce the error of the test process under the same conditions.
R3. The hydrothermal synthesis of all three samples were repeated many times and x-ray diffraction analysis revealed the same crystalline phases.
- Figure 5 and Figure 6 are not clear enough so it is recommended to optimize.
R4. The modifications are done. We changed all the images with other images that have 1200 DPI resolution.

Reviewer 2 Report
The authors of the article consider the use of defects in BiFeO3 semiconductors, which can provide a key technology to overcome undesirable limitations associated with both current leakage and other problems arising in perovskite structures. To characterize the results obtained, the authors used a large number of different methods of analysis, and also established new relationships. In general, this work is very interesting and promising, and also corresponds to the subject of the declared journal. However, before accepting the article, the authors should provide answers to all the questions of the reviewer, as well as make corrections to the text.
1. In the abstract, the authors point out that oxygen vacancies are responsible for the defect structure, in this regard, this statement requires more confirmation, as well as reference to other works related to similar studies.
2. The authors should provide more data on the possibilities of obtaining p-type BiFeO3 perovskites using hydrogen peroxide, as well as the conditions for their modification.
3. When describing the experimental part, the authors should indicate the chemical purity and manufacturers of chemical reagents.
4. The authors should indicate how exactly the phase composition of ceramics was determined, as well as provide calculation formulas or a description of the methodology for calculating the contributions of various phases.
5. The authors should provide more data on the conductive properties of ceramics, as well as compare the obtained data on conductive characteristics with the density values of ceramics.
6. Authors should provide data from morphological studies if possible.
7. Abstract and Conclusion should be expanded to introduce new data as well as provide further research perspectives.
Author Response
Thank you very much for the comments. We appreciate the referee’s time and effort to improve the manuscript. We revised the manuscript according to the referee’s comments. We hope that our responses to the comments and the changes in the text will be accepted by you.
Reviewer 2: The authors of the article consider the use of defects in BiFeO3 semiconductors, which can provide a key technology to overcome undesirable limitations associated with both current leakage and other problems arising in perovskite structures. To characterize the results obtained, the authors used a large number of different methods of analysis, and also established new relationships. In general, this work is very interesting and promising, and also corresponds to the subject of the declared journal. However, before accepting the article, the authors should provide answers to all the questions of the reviewer, as well as make corrections to the text.
- In the abstract, the authors point out that oxygen vacancies are responsible for the defect structure, in this regard, this statement requires more confirmation, as well as reference to other works related to similar studies.
R1. However, few studies on VBi in BFO thin films and ceramics have been published, especially from the perspective of first-principles density functional calculations [19,20]. In accordance with the theoretical calculations, the most stable state for VBi is the fully ionized one, VBi3−, obtained by release of their holes, which determined the p-type conduction. Thus, BiFeO3 acts as a p-type semiconductor with a high concentration of h+. As a result, the conductivity of BFO can be reduced by decreasing the concentration of VBi through two mechanisms, namely increasing the donor concentration or decreasing the acceptor concentration. To our knowledge, one study reported that the conductivity of p-type BFO decreased through the reduction of the concentration of VBi due to the doping of BFO nanofibers with Sn by the sol–gel electrospinning technique [13].
- The authors should provide more data on the possibilities of obtaining p-type BiFeO3 perovskites using hydrogen peroxide, as well as the conditions for their modification.
R2. Our study aimed only to demonstrate that hydrogen peroxide (H2O2) can act as the electron donor, compensating h+ generated by the unoccupied acceptor VBi, effectively diminishing the concentration of h+ and therefore leading to an increase in the electrical resistivity. In our future work, we will study the effect of the amount of hydrogen peroxide on Bi vacancies, reflecting on the dielectric and electrical properties.
- When describing the experimental part, the authors should indicate the chemical purity and manufacturers of chemical reagents.
R3. Separately, 1 mmol (0.5 g) of bismuth nitrate (Bi(NO3)3 × 5H2O ≥ 98%) and 1 mmol (0.4 g) of ferric nitrate (Fe(NO3)3 × 9H2O ≥ 98%) were mixed in 5 ml distilled water. These two solutions were then homogenized for 15 minutes using magnetic stirring, generating a brownish-yellow solution. The sample mixture was combined with 10 ml of 1M sodium hydroxide (Na (OH) ≥ 99%) solution before being transferred to a 60 ml Teflon line autoclave then heated at 200 °C for 12 hours.
- The authors should indicate how exactly the phase composition of ceramics was determined, as well as provide calculation formulas or a description of the methodology for calculating the contributions of various phases.
R4. The phase composition of the ceramics was determined using the X'Pert HighScore Plus software.
- The authors should provide more data on the conductive properties of ceramics, as well as compare the obtained data on conductive characteristics with the density values of ceramics.
R5. We calculated the relative densities of all samples rule out any effect of disk preparation on the electrical studies, and introduced two paragraphs in section 2 and section 3 respectively:
“These disks were weighted, measured, and densities were determined using Archimedes' technique, after sintering. The ceramics' relative densities were calculated as a percentage of the theoretical density.”
“The characteristics of BFO ceramics are heavily influenced by factors like grain size and density. The observed results suggest that the experimental conditions used in the synthesis are suitable for the production of highly dense ceramics. Sample 2 has the highest degree of densification (relative density of 93%), while samples 1 and 3 have comparable values (≈92%). The high relative densities of all samples rule out any effect of disk preparation on the electrical studies.”
- Authors should provide data from morphological studies if possible.
R6. SEM images (Fig.2a, 2b and 2c) reveal that the resultant BFOs are large-scale aggregation of truncated and highly deformed polyhedra with an average edge length of roughly 10 µm. Moreover, no difference in the size or morphology of the produced BFO ceramics by hydrothermal route with and without hydrogen peroxide was observed.
- Abstract and Conclusion should be expanded to introduce new data as well as provide further research perspectives.
R7. Enhancing the dielectric and electrical properties achieved by the H2O2-synthesised BFO ceramic confirms that bismuth vacancies can be reduced by this method. In our future work, we will study the effect of the amount of hydrogen peroxide on Bi vacancies, reflecting on the dielectric and electrical properties. In addition, the hydrogen peroxide-assisted hydrothermal synthesis might be extended to the synthesis of other perovskite materials.

Reviewer 3 Report
The paper entitled „Increasing Electrical Resistivity of p-type BiFeO3 Ceramics by Hydrogen Peroxide-Assisted Hydrothermal Synthesis” focuses on the reduction of the concentration of Bi vacancies during the hydrothermal synthesis of Bismuth ferrite (BFO) ceramics by adding hydrogen peroxide in the solution. The ceramic powders have been characterized by XRD analysis, Fourier transform infrared (FT-IR) spectroscopy, electrochemical and dielectric measurements. The paper is interesting and well-explained. Although the introduction refers to the aim of the study and the results are understandably submitted and sufficiently illustrated, the conception of the study should be changed. The influence of the changing hydrogen peroxide concentration should be revealed.
I would like to recommend the publication of the paper publication after some changes concerning the following issues:
1. More quantitative findings about the experimental results should be included in the abstract.
2. The abbreviation of the samples, together with their production process specifics could be better presented if tabulated in section 2. Materials and Methods.
3. The concept of the study should be changed. Why S2 sample with a tiny amount of bismuth nitrate was examined in this study and no variation of the quantity of H2O2/H2O was proposed? It follows that there are two reference samples and only one synthesized with hydrogen peroxide. How did the authors choose exactly a 5 ml to 10 ml ratio of H2O2/H2O in sample 3? A gradual change in the concentration of hydrogen peroxide should be presented and more samples produced with various hydrogen peroxide concentrations should be included in this study.
4. In Figure 1, it is not clear which sample is indicated with a circle sign.
5. In Figure 2, it would be better to indicate with numbers the position of the absorption peak.
6. Is there a difference in the size or morphology of the produced powder materials by hydrothermal route with and without hydrogen peroxide? If a difference is observed, this could contribute to the examined properties' variation.
Author Response
Thank you very much for the comments. We appreciate the referee’s time and effort to improve the manuscript. We revised the manuscript according to the referee’s comments. We hope that our responses to the comments and the changes in the text will be accepted by you.
Reviewer 3: The paper entitled „Increasing Electrical Resistivity of p-type BiFeO3 Ceramics by Hydrogen Peroxide-Assisted Hydrothermal Synthesis” focuses on the reduction of the concentration of Bi vacancies during the hydrothermal synthesis of Bismuth ferrite (BFO) ceramics by adding hydrogen peroxide in the solution. The ceramic powders have been characterized by XRD analysis, Fourier transform infrared (FT-IR) spectroscopy, electrochemical and dielectric measurements. The paper is interesting and well-explained. Although the introduction refers to the aim of the study and the results are understandably submitted and sufficiently illustrated, the conception of the study should be changed. The influence of the changing hydrogen peroxide concentration should be revealed.
I would like to recommend the publication of the paper publication after some changes concerning the following issues:
- More quantitative findings about the experimental results should be included in the abstract.
R1. The reduction of Bi vacancies highlighted by FT-IR and Mott-Schottky analysis has an expected contribution to the dielectric characteristic, decreasing the dielectric constant (with approximately 40%) and loss (3 times) together with the increase of the electrical resistivity (3 times) achieved by the hydrogen peroxide-assisted hydrothermal synthesized BFO ceramic compared to hydrothermal synthesized BFO.
- The abbreviation of the samples, together with their production process specifics could be better presented if tabulated in section 2. Materials and Methods.
R2. The corection was made in section 2.
- The concept of the study should be changed. Why S2 sample with a tiny amount of bismuth nitrate was examined in this study and no variation of the quantity of H2O2/H2O was proposed? It follows that there are two reference samples and only one synthesized with hydrogen peroxide. How did the authors choose exactly a 5 ml to 10 ml ratio of H2O2/H2O in sample 3? A gradual change in the concentration of hydrogen peroxideshould be presented and more samples produced with various hydrogen peroxide concentrations should be included in this study.
R3. For lower values of the ratio of H2O2/H2O, no changes of the crystalline phase were observed. The first modification of the crystalline structure accompanied by changes in the dielectric and electrical properties was highlighted for a 5 ml to 10 ml ratio of H2O2/H2O. Our study aimed only to demonstrate that hydrogen peroxide (H2O2) can act as the electron donor, compensating h+ generated by the unoccupied acceptor VBi, effectively diminishing the concentration of h+ and therefore leading to an increase in the electrical resistivity. In our future study, we will study the effect of hydrogen peroxide (increasing the ratio of H2O2/H2O taking into account that the pressure inside the autoclave should not exceed 50 bars) on Bi vacancies and how to change the dielectric and electrical properties.
- In Figure 1, it is not clear which sample is indicated with a circle sign.
R4. Figure 1 was remade.
- In Figure 2, it would be better to indicate with numbers the position of the absorption peak.
R5. Figure 2 was edited according to reviewer’s comment.
- Is there a difference in the size or morphology of the produced powder materials by hydrothermal route with and without hydrogen peroxide? If a difference is observed, this could contribute to the examined properties' variation.
R6. SEM images (Fig.2a, 2b and 2c) reveal that the resultant BFOs are large-scale aggregation of truncated and highly deformed polyhedra with an average edge length of roughly 10 µm. Moreover, no difference in the size or morphology of the produced BFO ceramics by hydrothermal route with and without hydrogen peroxide was observed.

Reviewer 4 Report
In this work entitled “Increasing Electrical Resistivity of p-type BiFeO3 Ceramics by Hydrogen Peroxide-Assisted Hydrothermal Synthesis”, BiFeO3 ceramics with high electrical resistivity have been obtained using hydrogen peroxide (H2O2) as part of hydrothermal medium. The crystal structure was characterized by XRD studies. FT-IR analysis and Mott-Schottky studies were performed, and the dielectric behavior and electrical resistance were studied. However, the work offers little advancement for improving the dielectric and resistivity properties of BiFeO3 material. Thus, the contribution of this paper is limited. In addition, the properties of BiFeO3 including dielectric and resistivity properties were well studied in the literature, and the resistivity obtained in this work is much lower than the values reported in literature, for instance:
1. G.S. Dias, I.B. Catellani, L.F. Cotica, I.A. Santos, V.F. Freitas, F. Yokaichiya, Highly resistive fast-sintered BiFeO3 ceramics, Integrated Ferroelectrics 174 (2016) 43-49.
2. N. Wang, X. Luo, L. Han, Z. Zhang, R. Zhang, H. Olin, Y. Yang, Structure, Performance, and Application of BiFeO3 Nanomaterials, Nano-Micro Letters 12 (2020) 81.
3. Wang, Yu-Fei and Xu, Chunxin and yan, long and Qiao, Xiaoshuang, Precise Adjustment of Forbidden Bandwidth in BiFeO3/Bi25FeO40 Heterojunction Structure. Available at SSRN: https://ssrn.com/abstract=4354305 or http://dx.doi.org/10.2139/ssrn.4354305
In addition, there are some suggestions for a further improvement of the manuscript which the authors could consider.
The XRD data in Figure 1 should be refined, such as using GSAS program to determine the phase, and the cell parameters could be given by the authors.
How did the authors exactly qualify the percentage of BiFeO3 / Bi25FeO40 of the composite in Table1. Is it possible to precisely control the percentage and is it repeatable?
The morphology of the obtained materials could be of interest for the readers of the journal of Materials, such as using SEM or STEM studies to characterize the microstructure of prepared materials.
The manuscript should be carefully rechecked, and extensive editing of English language and style is required throughout the manuscript due to too many mistakes.
Author Response
Thank you very much for the comments. We appreciate the referee’s time and effort to improve the manuscript. We revised the manuscript according to the referee’s comments. We hope that our responses to the comments and the changes in the text will be accepted by you.
Reviewer 4: In this work entitled “Increasing Electrical Resistivity of p-type BiFeO3 Ceramics by Hydrogen Peroxide-Assisted Hydrothermal Synthesis”, BiFeO3 ceramics with high electrical resistivity have been obtained using hydrogen peroxide (H2O2) as part of hydrothermal medium. The crystal structure was characterized by XRD studies. FT-IR analysis and Mott-Schottky studies were performed, and the dielectric behavior and electrical resistance were studied. However, the work offers little advancement for improving the dielectric and resistivity properties of BiFeO3 material. Thus, the contribution of this paper is limited.
- In addition, the properties of BiFeO3 including dielectric and resistivity properties were well studied in the literature, and the resistivity obtained in this work is much lower than the values reported in literature, for instance:
- G.S. Dias, I.B. Catellani, L.F. Cotica, I.A. Santos, V.F. Freitas, F. Yokaichiya, Highly resistive fast-sintered BiFeO3 ceramics, Integrated Ferroelectrics 174 (2016) 43-49.
- N. Wang, X. Luo, L. Han, Z. Zhang, R. Zhang, H. Olin, Y. Yang, Structure, Performance, and Application of BiFeO3 Nanomaterials, Nano-Micro Letters 12 (2020) 81.
- Wang, Yu-Fei and Xu, Chunxin and yan, long and Qiao, Xiaoshuang, Precise Adjustment of Forbidden Bandwidth in BiFeO3 / Bi25FeO40 Heterojunction Structure. Available at SSRN: https://ssrn.com/abstract=4354305or http://dx.doi.org/10.2139/ssrn.4354305
R1. Indeed, the values of the resistivity reported in literature mentioned by the reviewer are high, but these papers studied the n-type BiFeO3. In our study, the resistivity of p-type BiFeO3 ceramics was firstly increased by hydrogen peroxide-assisted hydrothermal synthesis So far, few studies on VBi in p-type BFO thin films and ceramics have been published especially from the perspective of first-principles density functional calculations [19,20]. To our knowledge, one study reported that the conductivity of p-type BFO decreased through the reduction of the concentration of VBi due to the doping of BFO nanofibers with Sn by the sol–gel electrospinning technique [13]
In addition, there are some suggestions for a further improvement of the manuscript which the authors could consider.
- The XRD data in Figure 1 should be refined, such as using GSAS program to determine the phase, and the cell parameters could be given by the authors.
R2. According to first principles density functional theory calculations [11], the presence of Bi vacancies will change the structural parameters and will be responsible for crystal volume reduction. The quantitative XRD data analysis results using Rietveld refinement with X’Pert HighScore Plus revealed that the crystal volume of S3 (373.75 Å) is higher than S1 (373.39 Å) and can be correlated with the reduction of Bi vacancies by using hydrogen peroxide in the hydrothermal synthesis.
- How did the authors exactly qualify the percentage of BiFeO3 / Bi25FeO40 of the composite in Table1. Is it possible to precisely control the percentage and is it repeatable?
R3. The phase composition of the ceramics was determined using the X'Pert HighScore Plus software. The hydrothermal synthesis of all three samples were repeated many times and x-ray diffraction analysis revealed the percentage of the crystalline phases.
- The morphology of the obtained materials could be of interest for the readers of the journal of Materials, such as using SEM or STEM studies to characterize the microstructure of prepared materials.
R4. SEM images (Fig.2a, 2b and 2c) reveal that the resultant BFOs are large-scale aggregation of truncated and highly deformed polyhedra with an average edge length of roughly 10 µm. Moreover, no difference in the size or morphology of the produced BFO ceramics by hydrothermal route with and without hydrogen peroxide was observed.
- The manuscript should be carefully rechecked, and extensive editing of English language and style is required throughout the manuscript due to too many mistakes.
R5. The corrections were done.

Round 2
Reviewer 2 Report
The authors answered all the questions, the article can be accepted for publication.
Author Response
Thank you very much for your acceptance
Reviewer 3 Report
The authors have carefully addressed the reviewer's recommendations.
Author Response
Thank you very much for your acceptance
Reviewer 4 Report
As the authors have refined the XRD data, the refinement can also be presented in Figure 1, and a table including the refinenment results, cell parameters could be given by the authors.
Author Response
Thank you very much for the comments. The refinement was presented in a new figure (figure 2) and the unit cell parameters were introduced in table 1.